# Effects of Castration on miRNA, lncRNA, and mRNA Profiles in Mice Thymus

**DOI:** 10.3390/genes11020147

**Published:** 2020-01-30

**Authors:** Bingxin Li, Kaizhao Zhang, Yaqiong Ye, Jingjing Xing, Yingying Wu, Yongjiang Ma, Yugu Li

**Affiliations:** 1College of Veterinary Medicine, South China Agricultural University, Guangzhou 510642, China; libingxin212@126.com (B.L.); zkz12320092006@163.com (K.Z.); xingjingjingda@126.com (J.X.); 18437958513@163.com (Y.W.); mayongjiang@scau.edu.cn (Y.M.); 2School of Life Science and Engineering, Foshan University, Foshan 528000, China; cn874462@163.com

**Keywords:** ovariectomy, orchiectomy, mice, thymus, miRNA, lncRNA

## Abstract

Thymic degeneration and regeneration are regulated by estrogen and androgen. Recent studies have found that long non-coding RNAs (lncRNAs) and microRNAs (miRNAs) are involved in organ development. In this study, RNA sequencing (RNA-seq) results showed that ovariectomy significantly affected 333 lncRNAs, 51 miRNAs, and 144 mRNAs levels (*p* < 0.05 and |log2fold change| > 1), and orchiectomy significantly affected 165 lncRNAs, 165 miRNAs, and 208 mRNA levels in the thymus. Gene Ontology (GO) and Kyoto Encyclopedia of Genes and Genomes (KEGG) analysis showed that differentially expressed genes (DEGs) were closely related to cell development and immunity. Next, we constructed two lncRNA–miRNA–mRNA networks using Cytoscape based on the targeting relationship between differentially expressed miRNAs (DEMs) and DEGs and differentially expressed lncRNAs (DELs) analyzed by TargetScan and miRanda. Besides, we screened DEGs that were significantly enriched in GO and in ceRNA networks to verify their expression in thymocytes and thymic epithelial cells (TECs). In addition, we analyzed the promoter sequences of DEGs, and identified 25 causal transcription factors. Finally, we constructed transcription factor-miRNA-joint target gene networks. In conclusion, this study reveals the effects of estrogen and androgen on the expression of miRNAs, lncRNAs, and mRNAs in mice thymus, providing new insights into the regulation of thymic development by gonadal hormones and non-coding RNAs.

## 1. Introduction

As a primary immune organ, the thymus regulates adaptive immunity by producing naïve T lymphocytes [1]. Thymus degeneration is closely related to age, and the main age-related manifestations are thymus size becomes smaller, reduced thymic epithelial cells, and gradually decreased naive T lymphocyte output [2,3]. When naive T lymphocytes in the peripheral circulation system decrease with aging, effector-memory T lymphocytes will proliferate to maintain a constant number of total lymphocytes [4,5,6]. However, the reactivity of old lymphocytes to new infections is weak. Therefore, the reduction in the number of naïve T lymphocytes is considered to be one of the causes of the decline in resistance to infections and cancer in older adults [7,8]. Moreover, Gui found that the number of thymocytes in female and male mice decreased sharply at 3 months of age [9].

Gonadal hormone levels are also closely related to aging [10]. Estrogen and androgen can regulate thymus development, which has been widely reported [11,12]. Studies have found that pregnancy or estrogen injection can induce thymus atrophy and inhibit the expression of chemokines, such as monocyte chemoattractant protein-1 and stromal cell-derived factor 1 [13,14]. Chemokines are essential for thymocyte migration into the thymus and induce T lymphocyte differentiation [15]. In addition, androgen therapy can also induce thymus atrophy [11]. Heng et al. found that prepubertal castration significantly reduced the rate of thymic degeneration and increased early T lineage progenitors [16]. Interestingly, the thymus rapidly regenerated when ovaries or testicles were removed [17,18]. Therefore, exploring the effects of estrogen and androgen on thymic development is very valuable.

microRNAs (miRNAs) are conserved endogenous 18–26 nucleotide (nt) RNAs that can target the 3’UTR gene region to regulate gene expression [19]. In animal genomes, miRNAs are involved in regulating more than 30% of gene expression [20]. Recent studies have found that miRNAs widely participate in the regulation of thymus aging in mice [2,21]. Our previous study found that miR-195a-5p can inhibit the proliferation of thymic epithelial cells (TECs) by targeting Smad family member 7 [22]. In addition, we also found that miRNA181a-5p may enhance TECS proliferation by regulating TGF-β signal transduction [23]. Although some studies have been performed on the effects of miRNAs on thymic aging, additional miRNAs and their regulatory mechanisms require extensive investigation.

Long non-coding RNAs (lncRNAs) are important components of non-coding RNAs [24]. lncRNAs can regulate the expression of target genes in a variety of ways, such as cis, trans, or combined with miRNAs, so it is not easy to determine their target genes [25,26]. The roles of lncRNAs in organ development, cancer treatment, and aging have been widely reported over the past decades [27,28,29]. In addition, Hu et al. found that lncRNAs play an important role in the differentiation of T helper subsets [30]; Li et al. found that the levels of 17β-estradiol (E2) and progesterone in mice ovarian cells increased after overexpression of lncRNA SRA [24]. A recent study found that androgen and estrogen influence lncRNA expression during penis development in marsupials [31]. Meanwhile, our previous study found that E2 treatment can inhibit cell activity and proliferation, and influence the lncRNA expression profile in TECS [29]. However, the effects of gonadal hormones on RNAs in the thymus have not been reported. In this study, we removed mice ovaries and testes to explore the effects of estrogen and androgen on lncRNA, miRNA, and mRNA expression profiles in the thymus. Subsequently, functional analysis and annotation of differentially expressed genes (DEGs) were performed to further explore the importance of gonadal hormones in thymic degeneration. Finally, we constructed and preliminarily validated the lncRNA–miRNA–messenger RNA (mRNA) regulatory network to determine the key factors of estrogen and androgen regulation of thymic degeneration.

## 2. Materials and Methods

### 2.1. Ethics Approval

All the methods in this experiment are based on animal science experimental guidelines. In addition, all animal experiment programs were approved by the Animal Protection Committee of South China Agricultural University (Guangzhou, China), with approval number SCAU 0014.

### 2.2. Animals and Sample Collection

For the experiment, 48 one-month-old CD1 mice (half male and half female) were purchased from the Laboratory Animal Center of Guangzhou University of Chinese Medicine (license key: SCXK (Yue) 2013-0034). The mice were fed in a specific pathogen-free environment (12/12 h light/dark cycle, 22–24 °C, 40–60% humidity) and randomly subdivided into ovariectomy group (F3x), female control group (F3), orchiectomy group (M3x), and male control group (M3) (12 mice per group). Mice were allowed to adapt for 7 days before the experiment. On the eighth day, the mice in the treatment groups were anesthetized for ovary or testicle removal, and the control groups received the same treatment but were not neutered. At 3 months old [9], the thymuses of each group were collected aseptically, frozen immediately with liquid nitrogen, and stored at –80 °C until use.

### 2.3. Total RNA Extraction and Sequencing

Referring to known research and economic factors, we mixed the tissues of each group equally for sequencing by a service provider (LC-BIO Bio-tech ltd, Hangzhou, China). According to the manufacturer’s instructions, total RNA was extracted from mice thymus using Trizol reagent (Invitrogen, Carlsbad, CA, USA). A Bioanalyzer 2100 (Agilent, Santa Clara, CA, USA) and agarose gel electrophoresis were used to assess the quantity, purity, and integrity of total RNA. Total RNA with RNA integrity number greater than 7 was selected for further experiments. Then a TruSeq Small RNA Sample Prep Kit (Illumina, San Diego, CA, USA) was used to construct miRNA libraries according to the manufacturer’s instructions. Finally, we performed single-ended sequencing on an Illumina HiSeq 2500 (Illumina, San Diego, CA, USA).

Ribosomal RNA (rRNA) was depleted from DNA-free RNA by an Epicentre Ribo-Zero Gold Kit (Illumina, San Diego, CA, USA). Then rRNA-free RNA was reverse-transcribed to create the final complementary DNA (cDNA) libraries by mRNA-Seq sample preparation kit (Illumina, San Diego, CA, USA). Finally, the libraries were paired-end sequenced on an Illumina HiSeq 4000 (Illumina, San Diego, CA, USA).

### 2.4. Transcript Assembly

Cutadapt [32] and FastQC (http://www.bioinformatics.babraham.ac.uk/projects/fastqc/) were used to remove reads containing adaptors and low-quality bases, and to verify sequence quality (Figure 1). Then clean data were mapped to the mice genome using Bowtie 2 [33] and Tophat 2 [34]. Finally, StringTie [35] and Ballgown [36] were used to merge all transcriptomes from this experiment and estimate their expression levels by calculating fragments per kilobase per million reads.

### 2.5. miRNA Identification

The raw data were processed using the ACGT101-miR internal program (LC Sciences, Houston, TX, USA) to remove adaptor dimers, garbage, low complexity, common RNA families (rRNA, tRNA, snRNA, snoRNA), and repeats. Then we used miRBase 21.0 to define known miRNAs. Finally, unmapped sequences were predicted using RNAfold software (http://rna.tbi.univie.ac.at/cgi-bin/RNAfold.cgi).

### 2.6. lncRNA Identification

First, we removed all transcripts that overlapped with known mRNAs and were shorter than 200 bp. Then CPC [37], CNCI [38], and Pfam [39] were used to predict transcripts with coding potential. All transcripts with CPC score <–1 and CNCI score <0 were deleted. Finally, the transcripts annotated as i, j, o, u, and x were retained. These letters represent transfrags falling entirely within a reference intron; potentially novel isoform; generic exonic overlap with a reference transcript; and unknown, intergenic, and exonic overlap with reference on the opposite strand, respectively.

### 2.7. RNA Expression and Functional Analysis

Differentially expressed lncRNAs (DELs), miRNAs (DEMs), and DEGs were screened by the R package Ballgown with log2 (fold change (FC)) > 1 or log 2 (FC) < –1 and statistical significance *p* < 0.05. Gene Ontology (GO) and Kyoto Encyclopedia of Genes and Genomes (KEGG) in the DAVID online tool (https://david.ncifcrf.gov/summary.jsp) were used to analyze the functions of DEGs; *p* < 0.05 represents significant difference.

### 2.8. lncRNA–miRNA–mRNA Network Analysis

Recent studies have shown that lncRNAs and mRNAs containing the same miRNA binding site can regulate each other’s expression levels by competitively binding miRNAs. To determine the interactions between DELs, DEMs, and DEGs after ovariectomy or orchiectomy, miRWalk, miRanda, RNAhybrid, and Targetscan were used for screening. Then, the intersections of co-expressed lncRNAs, miRNAs, and mRNAs in the 4 software programs were used to construct a lncRNA–miRNA–mRNA network. Finally, the lncRNA–miRNA–mRNA network was visualized using Cytoscape v3.7 software (https://cytoscape.org/). As competing endogenous RNA (ceRNA) analysis is based on miRNAs, the relationship between lncRNAs and mRNAs needs further investigation.

### 2.9. Cell Isolation and Culture

Thymocytes were isolated from 1-month-old mice. According to the manufacturer’s instructions, thymocytes were obtained by centrifugation using the lymphocyte separation solution (Dakewe, Shenzhen, China). Red blood cell lysate (Solarbio, Beijing, China) was used to remove red blood cells. Then, cells were diluted with RPMI-1640 (Gibco, Waltham, MA, USA) containing 10% fetal bovine serum (Gibco, Waltham, MA, USA) and 0.2% penicillin / streptomycin (Invitrogen, Carlsbad, CA, USA) and culture for 24 h in 37 °C at 5% CO_2_ to remove adherent cells. TECS were isolated from mice thymus in our previous experiments [40]. TECS were cultured in DMEM (Gibco, Waltham, MA, USA) containing 10% fetal bovine serum (Gibco, Waltham, MA, USA) and 0.2% penicillin / streptomycin (Invitrogen, Carlsbad, CA, USA) at 37 °C in 5% CO_2_. Finally, thymocytes and TECS were collected and immediately placed in liquid nitrogen and stored at −80 °C until use.

### 2.10. Quantitative Real-Time PCR Analysis

Total RNA in mice thymus, thymocytes and TECS were extracted with Trizol reagent (Takara, Kusatsu, Japan). First-strand cDNA was synthesized with a ReverTra Ace quantitative real time-PCR (qRT-PCR) RT Kit (Toyobo, Osaka, Japan), with random hexamers (for mRNAs and lncRNAs) and stem-loop RT primers (for miRNAs, purchased from RiboBio) according to the manufacturer’s instructions. Primers of mRNAs and lncRNAs (Appendix A) were designed by Primer Premier 5.0 software (Premier Biosoft International, USA), and miRNA primers were purchased from RiboBio (Guangzhou, China). β-actin (for mRNAs and lncRNAs) and U6 (for miRNAs) were used to normalize the data. qRT-PCR was performed by a Bio-Rad CFX96 Real-Time PCR system (Bio-Rad, Hercules, CA, USA) using SYBR Green Real-Time PCR Master Mix (Toyobo) following the manufacturer’s instructions. All qRT-PCR assays were conducted in triplicate, and the relative levels were measured in terms of threshold cycle (Ct) and calculated using the formula 2^−∆∆Ct^ [41].

### 2.11. Analysis of Transcription Factor Binding Sites of DEGs

The putative promoter sequence was obtained from Browser1 of the UCSC genome. TESS software v6.0 was used to searched Position-weigh matrices in the TRANSFAC database [42]. The relative fraction line was set to 0.9. Next, the internal PERL script was used to conduct hypergeometric testing. *p* < 0.05 was defined as enriched transcription factor.

### 2.12. Identification of Transcription Factor-Related miRNAs and Their Joint Target Genes

According to the analysis of DEGs-related transcription factors, we used CircuitsDB (http://biocluster.di.unito.it/circuits/) to obtain information about miRNAs and joint target genes related to these transcription factors. Cytoscape was used to build transcription factor-miRNAs-joint target genes networks

## 3. Results

### 3.1. Expression Profiles of miRNAs, lncRNAs, and mRNAs in Thymus

To identify putative transcripts in the thymus, five mice each from the F3x group, F3 group, M3x group, and M3 group were mixed together for sequencing. In this experiment, we obtained a total of 380 million uniquely mapped lncRNA reads (Table 1). In addition, we got 56,673 unique miRNA readings in the Rfam database (Table 2, Appendix A). In total, we identified 17,498 candidate lncRNAs, 1528 miRNAs, and 46,595 mRNAs in all chromosomes (Appendix A). Venny analysis showed that 13,343 lncRNAs, 878 miRNAs, and 37,643 mRNAs were expressed in all groups, accounting for 76.3%, 57.5%, and 80.8% of the total readings, respectively (Figure 2a–c). Chromosome analysis showed that lncRNAs were unevenly distributed on the chromosome and were basically consistent with the trends of chromosome length (Figure 2d,e). Interestingly, the minimum length, mean length, median length, and N50 of lncRNAs were very close to those of mRNAs in this experiment (Figure 2f, Table 3). In addition, the distribution of miRNAs showed that miRNA length was mainly concentrated in 20–24 nt (Figure 2g).

### 3.2. Identification of DELs, DEMs, and DEGs

Data analysis showed that ovariectomy significantly affected the levels of 333 lncRNAs, 51 miRNAs, and 144 mRNAs in mice thymus (*p* < 0.05, FC > 2 or FC < 0.5) (Figure 3a–c, Appendix A). Orchiectomy significantly affected the levels of 165 lncRNAs, 165 miRNAs, and 208 mRNAs (Figure 3d–f, Appendix A). To further explore the effects of gender on thymic degeneration, we analyzed differential RNAs in the F3x and M3x groups. The results showed that 416 lncRNAs, 68 miRNAs, and 225 mRNAs were remarkably differentially expressed after ovariectomy and orchiectomy (Appendix A, Appendix A). In addition, the levels of 136 lncRNAs, 129 miRNAs, and 124 mRNAs were remarkably different between the F3 and M3 groups (Appendix A, Appendix A). The analysis results show that most DEGs were upregulated after ovariectomy when compared with the F3 group, whereas most miRNAs and lncRNAs were downregulated (Figure 3a–c). It is worth noting that most DELs were upregulated after orchiectomy when compared with the M3 group (Figure 3d–f). Finally, 37 DELs, 10 DEMs, and 18 DEGs in the ovariectomy and female control groups and 27 DELs, 14 DEMs, and 36 DEGs in the orchiectomy and male control groups were used to construct lncRNA–miRNA–mRNA networks (Figure 3g–l).

In order to verify our RNA-seq data, we randomly selected 3 DEGs, 3 DELs, and 3 DEMs for qRT-PCR analysis (Figure 4). The results showed that the differentially expressed RNAs had the same expression trends in qRT-PCR and RNA-seq (R = 0.747, *p* = 0.003).

### 3.3. Functional and Pathway Annotation

GO terms and KEGG pathway annotations of DEGs were enriched via DAVID. The enriched GO terms were divided into 3 categories: biological process (BP), cellular component (CC), and molecular function (MF). In this study, the BP enrichment of DEGs after ovariectomy includes extracellular matrix organization (*p* = 0.015), peptide transport (*p* = 0.024), extracellular structure organization (*p* = 0.040), and cellular cation homeostasis (*p* = 0.042) (Figure 5a). KEGG enrichment of DEG after ovariectomy showed no significant enrichment pathway. Meanwhile, the BP enrichment of DEGs after orchiectomy includes immune response (*p* = 0.00000054), response to virus (*p* = 0.000025), defense response (*p* = 0.000036), inflammatory response (*p* = 0.0017), regulation of nitric oxide biosynthetic process (*p* = 0.0091), positive regulation of defense response (*p* = 0.0094), cellular alkene metabolic process (*p* = 0.011), positive regulation of response to stimulus (*p* = 0.014), production of nitric oxide during acute inflammatory response (*p* = 0.015). KEGG enrichment of DEG after orchiectomy includes arachidonic acid metabolism (*p* = 0.014), viral myocarditis (*p* = 0.020), and intestinal immune network for IgA production (*p* = 0.045).

### 3.4. lncRNA–miRNA–mRNA Network Construction and Visualization

As a member of ceRNAs, lncRNA can affect the expression of miRNA target genes by adsorbing miRNA. In this study, we screened co-expressed genes from differentially expressed RNAs to construct ceRNA networks. As shown in Figure 6a, the female lncRNA–miRNA–mRNA network included 37 lncRNAs, 10 miRNAs, and 18 mRNAs. The male lncRNA–miRNA–mRNA network included 27 lncRNAs, 14 miRNAs, and 36 mRNAs (Figure 6b). The results of the node degree distribution analysis in the ceRNAs network showed that the slope of the power law distribution in female was −1.115 and R^2^ = 0.723, and the slope in the male was -1.306 and R^2^ = 0.795, which indicates that these two ceRNA networks have typical bio-network scale-free feature (Figure 6c,d).

### 3.5. Expression of DEGs in Different Cells of Thymus

In order to further explore the function of DEGs in thymus development, we screened DEGs that were significantly enriched in GO and in the ceRNA network for verification. Then, qRT-PCR was used to detect the expression of these DEGs in thymocytes and TECS. As shown in Figure 7, H2-M2 was highly expressed in thymocytes. SLC7A2, CAV1, CACNB4, and PTGS2 were highly expressed in TECS. The expression trend of POLR3K in thymocytes and TECS was consistent.

### 3.6. Transcription Factors of DEGs

In order to explore the regulatory mechanism of DEGs, we used TESS software to predict enriched transcription factors of upregulated and downregulated DEGs. The results showed that the binding sites of KROX, Lmo2, PEA3, STAT1, and STAT4 were significantly overexpressed in the downregulated DEGs after ovariectomy (Figure 8a), and the binding sites of AP-1, AP-2rep, c-Myc, E2F-1 DP-2, ETF, MOVO-B, Pax, and ZF5 were significantly overexpressed in the upregulated DEGs (*p* < 0.05) (Figure 8b). Meanwhile, the binding sites of NF-kappaB, Pbx-1, SMAD, and SREBP-1 were significantly overexpressed in the downregulated DEGs after orchiectomy (Figure 8c), and the binding sites of Hoxa4, ICSBP, IRF, ISGF-3, MOVO-B, NF-γ, Oct-4, and TEF-1 were overexpressed in the upregulated DEGs (*p* < 0.05) (Figure 8d).

### 3.7. DEGs Transcription Factor-related DEMs and Joint Target Genes prediction

We use CircuitsDB to predict the transcription factors of miRNAs associated with DEGs, and predict the joint target genes. We found that one transcription factor (PEA3), which was related to DEGs after ovariectomy, was associated with mmu-miR-135a and 15 joint target genes (Figure 9a). In addition, we found that 5 DEGs-related transcription factors are associated with DEMs after orchiectomy. PBX-1 was associated with 3 DEMs, including mmu-miR-342, mmu-miR-135a, and mmu-miR-135b, and 5 joint target genes (Figure 9b). HOXA4 was associated with mmu-miR-135b and 2 joint target genes (Figure 9c). IRF was associated with mmu-miR-15a and 2 joint target genes (Figure 9d). SREBP-1 was associated with mmu-miR-200b and 3 joint target genes (Figure 9e). NF-γ was associated with 3 DEMs, including mmu-miR-342, mmu-miR-206, and mmu-miR-144, and 24 joint target genes (Figure 9f).

## 4. Discussion

The thymus, as a central immune organ, is the main site for generation of naive T lymphocytes [43]. Thymus atrophy is considered to be a biomarker of immune system aging and is influenced by many factors, such as age, hormones, and gender [44]. Studies have extensively demonstrated that ovariectomy or orchiectomy can result in transient thymic regeneration. However, the mechanisms by which gonadal hormones regulate thymic development remains unclear [17,18]. In addition, the role of non-coding RNA in regulating thymic development is revealed. In this study, 333 DELs, 51 DEMs, and 144 DEGs, and 165 DELs, 165 DEMs, and 208 DEGs were identified after ovariectomy and orchiectomy, respectively. In order to identify the accuracy of the RNA-seq data in this experiment, we randomly selected some differentially expressed RNAs for qRT-PCR identification. The results showed that DELs, DEMs, and DEGs had the same expression trends in qRT-PCR and RNA-seq (R = 0.747, *p* = 0.003), which parts supported the high reliability of our RNA seq data.

Next, GO and KEGG in DAVID were used to enrich the function of DEGs after ovariectomy and orchiectomy. DEGs after ovariectomy were mainly concentrated in extracellular structure organization, peptide transport, cellular cation homeostasis, and extracellular matrix organization in BP. Upregulation of peptide transport may help regulate lymphocyte proliferation [45]. Besides, cellular cation homeostasis is necessary for normal life activities of cells. For instance, K^+^ is used to compensate negative charges and activate protein translation processes [46]; Mg^2+^ is an important component of cell proliferation-related cofactors [47]. It is worth noting that the thymic extracellular matrix can increase thymocyte output in vivo and promote TEC differentiation in vitro [48]. Meanwhile, DEGs after orchiectomy were mainly concentrated in immune response and inflammatory response in BP. Castration or androgen deficiency may result in thymus enlargement, which may be related to androgen inhibiting immature thymocyte proliferation and accelerating its apoptosis [49]. In addition, androgens can affect the production of immune-related cytokines in the thymus. Upregulation of TGF-β and activation of TGF-β-induced Smad signaling pathway induces accumulation of connective fibers in the thymus [50]. As a proinflammatory cytokine, IL-6 can also stimulate the proliferation of thymocytes in vitro [51]. Besides, Oxidative stress affects the homeostasis of the immune system, causing the irreversibility of thymus degradation [52].

lncRNAs can participate in proliferation and aging physiological processes in different ways [53,54]. With the development of non-coding RNA research, the theory of ceRNA was proposed. In this theory, acting as key nodes, miRNAs can bind to lncRNAs to regulate the expressions of other non-coding RNAs or target genes with the same binding sites. In this experiment, we hypothesized that lncRNA is involved in regulating thymus development after mice castration through ceRNA. Next, we performed ceRNA analysis on the differentially expressed RNAs between castration and control groups and constructed two scale-free biological networks of ceRNAs. qRT-PCR preliminarily confirmed the expression of these genes in the ceRNA network were consistent with RNA-seq. In addition, the scale-free distribution of these two ceRNA networks indicates that there are important nodes in the regulation of gonadal hormones on thymus development.

In order to further clarify the function and expression site of DEGs in the ceRNA network. we screened DEGs (H2-M2, POLR3K, SLC7A2, CAV1, CACNB4, and PTGS2 that were significantly enriched in GO and in the ceRNA network to verify their expression in thymocytes and TECS. Cav1 is a channel for Ca2^+^ influx required for T cell development [55]. Cyclooxygenase 2 (COX-2) is encoded by the PTGS2 gene, which inhibits T cell proliferation by regulating NF-kappaB [56]. This experiment found that Cav1 and PTGS2 were significantly overexpressed in TECS, which may be related to the TEC providing a microenvironment for thymocyte development [23]. The qRT-PCR showed that H2-M2 and POLR3K were highly expressed in thymocytes, and SLC7A2 and CACNB4 were highly expressed in TECS. However, the roles of these genes in the thymus have not been reported. These genes may be worthy of further study.

In order to explore the regulatory mechanism of DEGs, we predicted the causal transcription factors of DEGs through enrichment experiments. We found that the binding sites of KROX, Lmo2, PEA3, STAT1, and STAT4 were significantly enriched in downregulated DEGs after ovariectomy, while AP-1, AP-2rep, c-Myc, E2F-1, DP-2, ETF, MOVO-B, Pax, and ZF5 binding sites were significantly enriched in upregulated DEGs. Meanwhile, the binding sites of NF-kappaB, Pbx-1, SMAD, and SREBP-1 were significantly enriched in downregulated DEGs after orchiectomy, while the binding sites of Hoxa4, ICSBP, IRF, ISGF-3, MOVO-B, NF-γ, Oct-4, and TEF-1 were significantly enriched in upregulated DEGs. The main function of Krox-20 is to inhibit the c-Jun NH2-terminal protein kinase-c-Jun pathway, whose activation is necessary for both proliferation and death [57]. Overexpression of LMO2 leads to delayed development of thymus progenitor cells and increases the number of T lymphocytes in lymphoid organs [58]. As members of the STAT family, TAT1 and STAT4 mainly promote the differentiation of Th1 cells [59]. AP-1 is one of the most important target members in the MAPK pathway, which can regulate cell division and apoptosis [60]. AP-2 rep is a transcriptional repressor of AP-2 and plays an important role in embryo development [61]. c-Myc makes cells sensitive to TNF stimulation and promotes apoptosis of thymocytes [62]. E2F-1 regulates apoptosis and proliferation by stabilizing p53 tumor suppressor [63]. The NF-kappaB pathway is considered to be a typical transduction pathway of proinflammatory cytokines represented by the IL-1 and TNF receptor families, which was required for T cell proliferation and emigration [64]. Smad is at the core of the TGF-β pathway [65]. HOXA4 expression is inversely related to cell cycle, metastasis, and Wnt signaling pathway, and its overexpression can inhibit tumor cell proliferation, migration and invasion [66]. Oct-4 belongs to the POU domain transcription factor family that are involved in regulating the proliferation and differentiation of various cells in tissues [67]. The function of some transcription factors in thymus development has been partially verified, but the role of PEA3, DP-2, ETF, MOVO-B, Pax, ZF5, Pbx-1, SREBP-1, ICSBP, IRF, ISGF-3, MOVO-B, NF-γ, and TEF-1 in regulating thymus development needs further investigation.

Based on the analysis of DEGs transcription factors, we used CircuitsDB to predict DEMs and associated target genes associated with them. The results showed that mmu-miR-135a was related to PEA3 after ovariectomy, and 7 DEMs (mmu-miR-342, mmu-miR-135a, mmu-miR-135b, mmu-miR-15a, mmu-miR-200b, mmu-miR-206, and mmu-miR-144) were associated with 5 DEGs-related transcription factors (PBX-1, HOXA4, IRF, SREBP, and NF-γ) after orchiectomy (Figure 9). miR-15a promotes the proliferation of mouse lymphocytes by regulating the expression of cell cycle-related genes [68]. miR-200b- 3p can be used as a biomarker for mouse thymus development and degradation [40]. These are consistent with our experimental results. However, studies on mmu-miR-342, mmu-miR-135a, mmu-miR-135b, mmu-miR-206, and mmu-miR-144 in mouse thymus have not been reported, so these genes deserve further investigation.

## 5. Conclusions

In summary, we identified the expression profiles of lncRNAs, miRNAs, and mRNAs in mice thymus from control groups and ovariectomy or orchiectomy group, and constructed two scale-free ceRNA networks. Bioinformatics analysis showed that DEGs were mainly enriched in extracellular matrix organization and immune response-related physiological functions. This study provides new insights into the effects of estrogen and androgen on thymic development, and the specific mechanisms need to be further explored.

## Figures and Tables

**Figure 1 genes-11-00147-f001:**
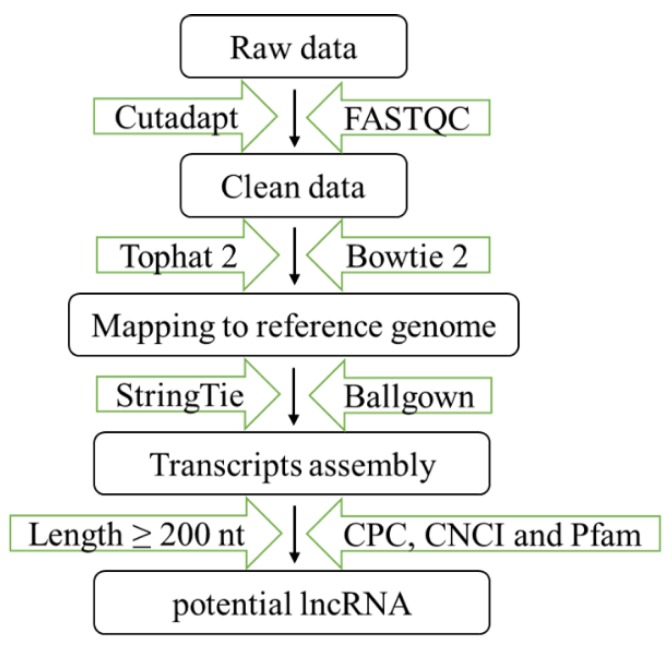
RNA library preparation and analysis process. CPC, Coding Potential Calculator; CNCI, Coding-Non-Coding Index.

**Figure 2 genes-11-00147-f002:**
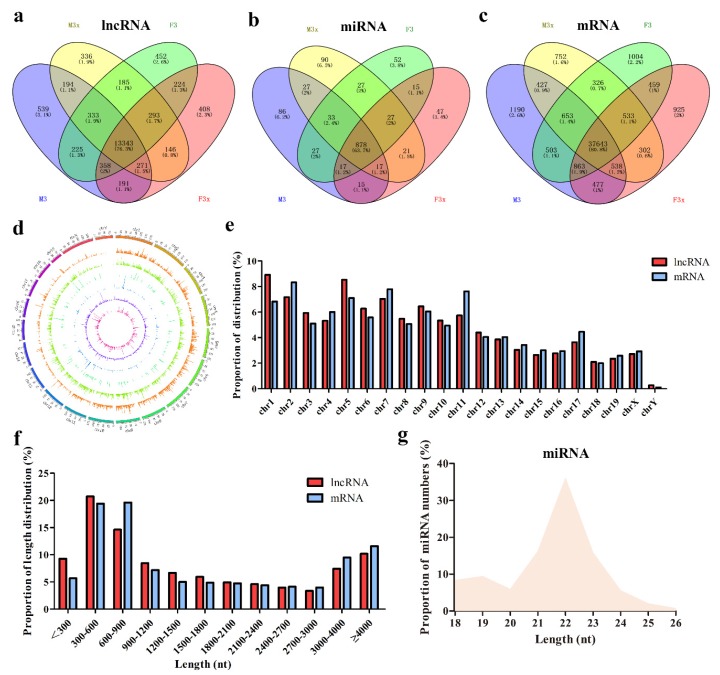
Identification of miRNAs, lncRNAs, and mRNAs in mice thymus. (**a–c**) Venny maps showing specific miRNAs, lncRNAs, and mRNAs, respectively, in the thymus after ovariectomy or orchiectomy. (**d**) Circos plot depicting the distribution of lncRNAs in different chromosomes. (**e**) Columnar graph showing the distribution of lncRNAs and mRNAs in different chromosomes. (**f**) The size and distribution of lncRNAs and mRNAs. (**g**) The distribution of miRNAs.

**Figure 3 genes-11-00147-f003:**
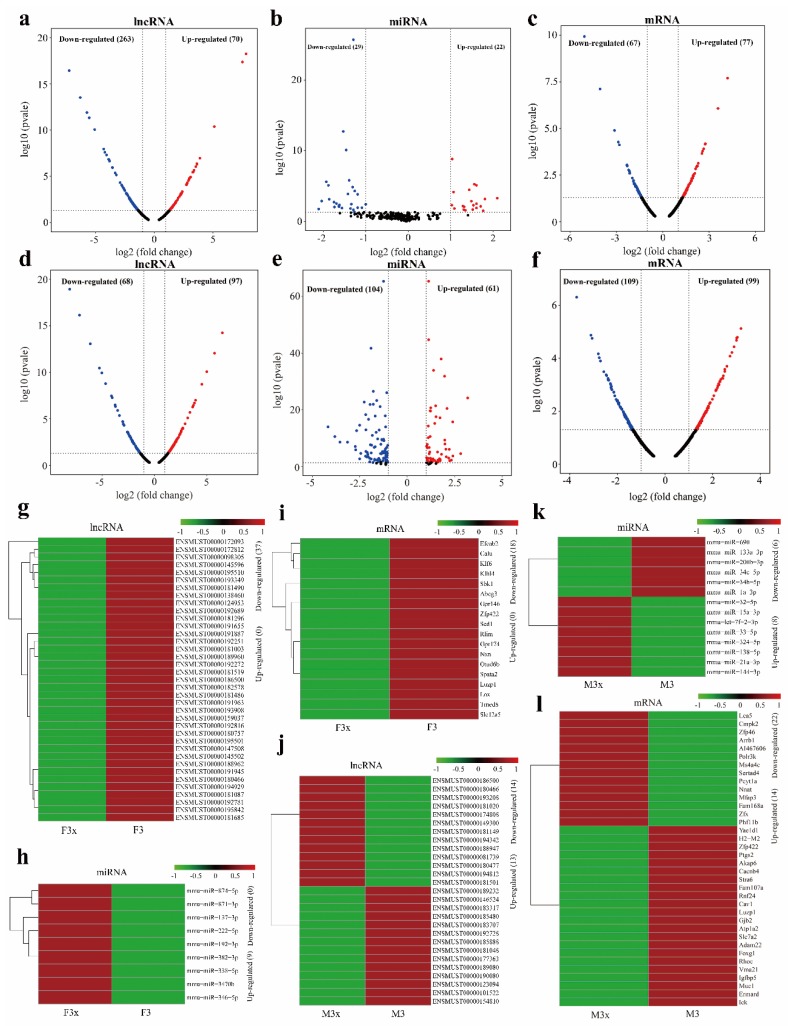
Comparative analysis of estrogen- and androgen-related RNAs in mice thymus. (**a**–**c**) Volcano plots of differentially expressed lncRNAs (DELs), differentially expressed miRNAs (DEMs), and differentially expressed genes (DEGs), respectively, in ovariectomy and female control groups. (**d–f**) Volcano plots of DELs, DEMs, and DEGs, respectively, in orchiectomy and male control groups. (**g–i**) Heat maps of specific DELs, DEMs, and DEGs, respectively, in ovariectomy and female control groups. (**j–l**) Heat maps of specific DELs, DEMs, and DEGs, respectively, in orchiectomy and male control groups. Each row represents one RNA. Green and red represent relatively lower and higher gene expression in castration group, respectively (*p* < 0.05 and |log2FC| > 1).

**Figure 4 genes-11-00147-f004:**
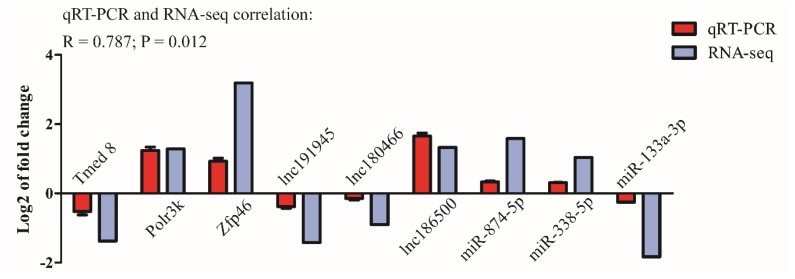
Validation of DELs, DEMs, and DEGs by qRT-PCR. Log2 fold change (FC) were expressed as mean ± SD. *n* = 3. The statistical significance of all genes reached *P* < 0.05.

**Figure 5 genes-11-00147-f005:**
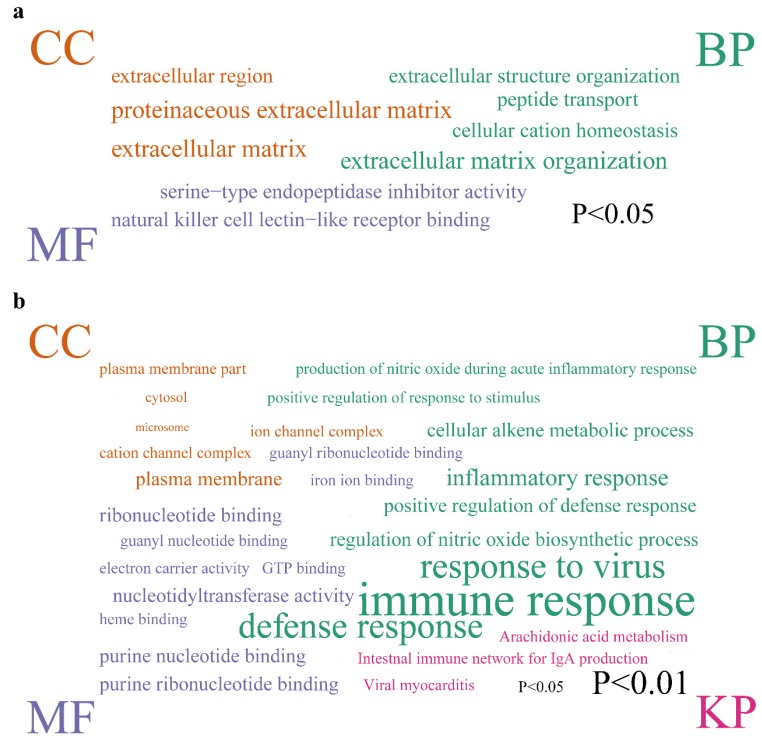
Functional analysis of significant DEGs in thymus of mice after ovariectomy or orchiectomy. Functional enrichment of DEGs after (**a**) ovariectomy and (**b**) orchiectomy. The significance cutoff for *P* value was set at 0.05. The font size of enriched terms is proportional to –log10 of *P* value.

**Figure 6 genes-11-00147-f006:**
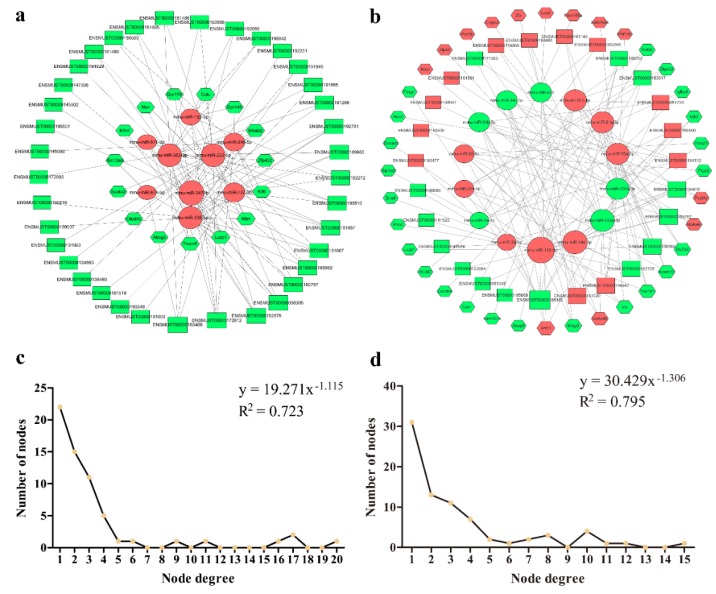
lncRNA–miRNA–mRNA network. (**a**) Global view of network after ovariectomy. (**b**) global view of network after orchiectomy. (**c**) Degree distribution of the female ceRNA network. (**d**) Degree distribution of the male ceRNA network. Rectangle, circle, and hexagon represents lncRNA, miRNA, and mRNA, respectively. Red represents upregulation of RNAs expression after castration, green represents downregulation.

**Figure 7 genes-11-00147-f007:**
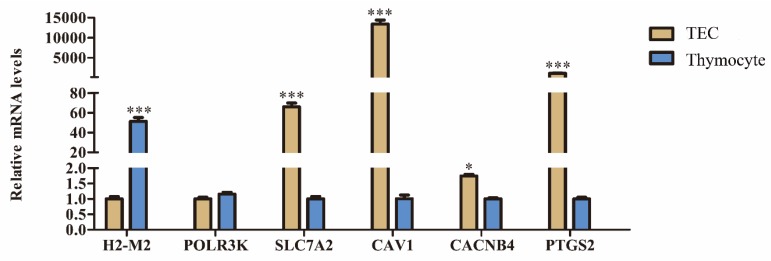
The expression levels of DEGs at thymic epithelial cells (TECS) and thymocytes. * represents *p* < 0.05, *** represents *p* < 0.001.

**Figure 8 genes-11-00147-f008:**
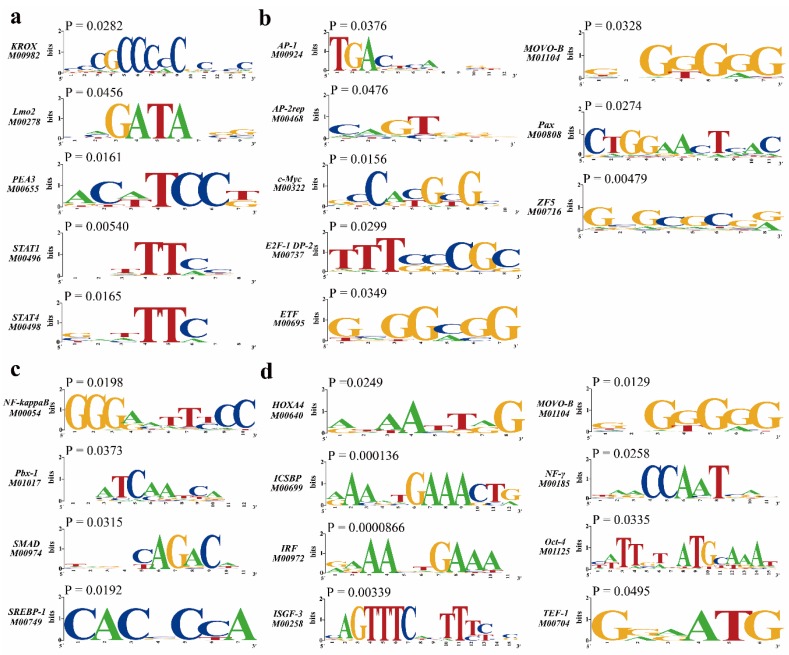
Analysis of transcription factor binding sites of DEGs. (**a**) Sequence logos of transcription factor binding sites in downregulated DEGs after ovariectomy. **(b**) Upregulated DEGs after ovariectomy. (**c**) Downregulated DEGs after orchiectomy. (**d**) Upregulated DEGs after orchiectomy.

**Figure 9 genes-11-00147-f009:**
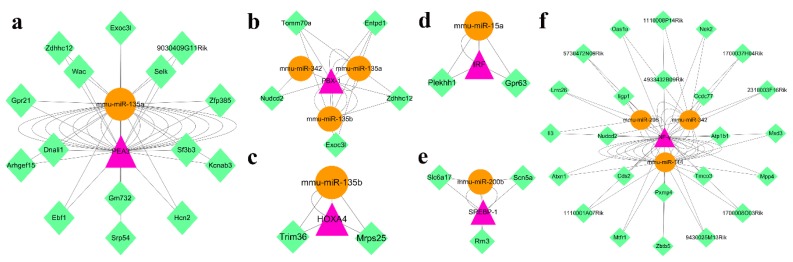
Transcription factor-miRNAs-joint target genes networks. (**a**) Transcription factor-miRNAs-joint target genes network after ovariectomy. (**b**–**f**) Transcription factor-miRNAs-joint target genes network after orchiectomy. The purple triangles, brown circles, and green diamonds represents transcription factors, miRNAs, and joint target genes, respectively.

**Table 1 genes-11-00147-t001:** Reading long non-coding RNAs (lncRNAs) mapped to reference genome.

Sample	M3	M3x	F3	F3x
Raw reads	147,021,844	148,097,166	146,835,736	154,631,520
Clean reads	138,324,292	137,140,672	138,697,620	144,375,220
Clean ratio (%)	94.08	92.60	94.46	93.37
Mapped reads	115,678,789	112,610,952	114,852,768	123,422,077
Mapping ratio (%)	91.35	91.33	91.54	91.26
Uniquely mapped reads	95,107,627	92,239,775	94,004,603	98,579,424
Unique mapping ratio (%)	75.10	74.81	74.93	72.89

**Table 2 genes-11-00147-t002:** Reading microRNAs (miRNAs) mapped to reference genome.

Group	Type	Raw Reads	3 ADT and Length Filter	Efam
M3	Total (%)	24,408,417 (100)	2,742,851 (11.24)	1,487,253 (6.09)
	Unique (%)	1,482,996 (100)	847,814 (57.17)	21,384 (1.44)
M3x	Total (%)	26,998,643 (100)	3,258,039 (12.07)	860,395 (3.19)
	Unique (%)	1,459,597 (100)	850,698 (58.28)	13,133 (0.90)
F3	Total (%)	22,244,392 (100)	2,661,777 (11.97)	703,157 (3.16)
	Unique (%)	1,283,499 (100)	779,232 (60.71)	12,022 (0.94)
F3x	Total (%)	20,753,214 (100)	2,332,868 (11.24)	590,578 (2.85)
	Unique (%)	1,131,578 (100)	635,854 (56.19)	10,134 (0.90)

**Table 3 genes-11-00147-t003:** Assembly results of lncRNAs and messenger RNAs (mRNAs).

Item	Min Length	Max Length	Mean Length	Median Length	N50
lncRNA	201	85,600	1972	1101	3320
mRNA	201	124,938	1984	1223	3246

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
