# Peer review of "Effects of Castration on miRNA, lncRNA, and mRNA Profiles in Mice Thymus"

_genes, 2020, doi:10.3390/genes11020147_

Round 1

Reviewer 1 Report

While significant effort has been made to improve the use of English and readability, there are still some grammatical errors in the Abstract that must be corrected.

Notably, the previous criticism about lack of information on functional and molecular effects of the identified lncRNAs, miRNAs, and mRNAs was addressed only by performing a litterature search for what is known about these molecules. There was no effort to establish their roles in the context of thymus development through serious laboratory experiments.

Author Response

Question 1: While significant effort has been made to improve the use of English and readability, there are still some grammatical errors in the Abstract that must be corrected.

Answer: Thank you for your suggestion. Your valuable suggestions have helped us a lot. Based on your suggestions, we have rigorously checked and corrected the Abstract. The detail was shown in Abstract in the manuscript.

Question 2: Notably, the previous criticism about lack of information on functional and molecular effects of the identified lncRNAs, miRNAs, and mRNAs was addressed only by performing a litterature search for what is known about these molecules. There was no effort to establish their roles in the context of thymus development through serious laboratory experiments.

Answer: Thank you for your suggestion. This is mainly related to our experimental purpose. Previous studies have found that gonadal hormones can affect thymic degeneration, but the mechanism is unclear. Meanwhile, noncoding RNAs are involved in organ development has been demonstrated. Therefore, the main purpose of this experiment is to investigate whether castration can affect the expression profile of non-coding RNAs in the thymus and screen noncoding RNAs regulated by gonadal hormones, so as to lay the foundation for further studying the molecular mechanism of gonadal hormones-induced thymic degradation. In this experiment, we analyzed the differentially expressed RNAs after castration and found that they were related to thymic development and immunity, which indicates that gonadal hormones may regulate thymic degradation through noncoding RNAs. The roles of candidate noncoding RNAs on thymic degradation is the focus of our next research.

Reviewer 2 Report

Yes I think that putting less emphasis on the two non-coding RNAs has made the paper better balanced. There is a typo in the second sentence of the abstract. 

Author Response

Question: Yes I think that putting less emphasis on the two non-coding RNAs has made the paper better balanced. There is a typo in the second sentence of the abstract.

Answer: Thank you for your suggestion. Your valuable suggestions have helped us a lot. We have changed the second sentence of the abstract 'Recent studies found that long noncoding RNAs (lncRNAs) and microRNAs (miRNAs) involuted in organs development' to 'Recent studies have found that long noncoding RNAs (lncRNAs) and microRNAs (miRNAs) are involved in organs development'. The detail was shown in Abstract in the manuscript.

This manuscript is a resubmission of an earlier submission. The following is a list of the peer review reports and author responses from that submission.

Round 1

Reviewer 1 Report

The paper is very well written and the results are sensibly analysed and interpreted.

There seems to be less of a difference between control and treated mice when qRT-PCR was used to confirm the sequencing results (Fig 6).   This does raise questions about the reliability of the sequencing data.   Do you have qRT-PCR data to confirm any of the most extreme read differences between the control and treated mice that you present in the supplementary data? Do you have reads for each mouse and if so are the biggest differences between groups also seen consistently between the individual mice in each group?

The biggest effect in Fig 6d was an increase in expression of ENSMUST00000186500 following testes removal. Is this lncRNA conserved across species?  What is known about it?   Are the other lncRNAs conserved?  Is there anything in the literature that adds support to your view that the mRNA, miRNA and lncRNA of Fig 6 are important for T cell regulation by gonads?

In the results I think you should explain why you assayed the thymus about 11 weeks after removing the gonads.   Perhaps cite the literature in order to explain what you would expect to have happened to the thymus or T cells at this point, or do you have any data?  Did your choice of 3 months have anything to do with reference 9 which you describe in the Introduction?  Alternatively, did you choose to analyse the thymus at a relatively early time point in order to try and detect key early events? 

Reviewer 2 Report

Quality of English is low. Article has many grammatical errors that impede reading and understanding what was written.

The functional and molecular implications/significance of the lncRNAs and miRNAs upregulated or downregulated are unclear.

The implication of ovariectomy and castration affecting lncRNAs, miRNAs, mRNAs, and ceRNA networks is not clear.

It is not clear if any of the findings have any potential applications in health/medicine.

There is no effort to determine the molecular mechanisms by which specific lncRNAs and miRNAs affect T lymphocyte production, and the functional implications of this.